# A Novel Method for Preparing Lightweight and High-Strength Ceramisite Coarse Aggregates from Solid Waste Materials

**DOI:** 10.3390/ma17112613

**Published:** 2024-05-29

**Authors:** Xin Xiong, Zhi Wu, Pengcheng Jiang, Min Lai, Guanghai Cheng

**Affiliations:** School of Materials Science and Engineering, Hunan Institute of Technology, No. 18 Henghua Road, Hengyang 421002, China; xwustx@163.com (X.X.); wustjpc@163.com (P.J.); lm526938378@163.com (M.L.); 15737620632@163.com (G.C.)

**Keywords:** ceramisite coarse aggregates, pore-forming method, coal ash floating beads, resource utilization of solid waste, fly ash

## Abstract

A novel method is introduced in this study for producing ceramisite coarse aggregates that are both lightweight and possess high strength. The process involves utilizing fly ash as the primary material, along with coal ash floating beads (CAFBs) that have high softening temperature and a spherical hollow structure serving as the template for forming pores. This study examined the impact of varying particle size and quantity of floating beads on the composition and characteristics of ceramisite aggregates. Results showed that the high softening temperature of floating beads provided stability to the spherical cavity structure throughout the sintering process. Furthermore, the pore structure could be effectively tailored by manipulating the size and quantity of the floating beads in the manufacturing procedure. The obtained ceramisite aggregates feature a compact outer shell and a cellular inner core with uniformly distributed pores that are isolated from each other and mostly spherical in form. They achieve a low density ranging from 723 to 855 kg/m^3^, a high cylinder compressive strength between 8.7 and 13.5 MPa, and minimal water absorption rates of 3.00 to 4.09%. The performance metrics of these coarse aggregates significantly exceeded the parameters specified in GB/T 17431.1-2010 standards.

## 1. Introduction

Ceramisite, an artificial lightweight aggregate, offers several advantages such as low apparent density, low thermal conductivity, and high durability when compared to traditional sand and gravel aggregates [1,2]. Concrete materials prepared with ceramisite as the aggregate have a lower density, better insulation properties, and simultaneously possess higher strength [2,3]. As a result, it holds promise for use in energy-saving and super high-rise buildings, as well as large-span bridges. However, the construction industry currently faces challenges related to high cost, significant performance fluctuations, and the low production yield of ceramisite used in construction. These issues significantly impede the widespread adoption and utilization of lightweight and high-strength concrete.

Currently, the high-temperature foaming method is the dominant technique utilized globally for producing lightweight and high-strength ceramisite aggregates. During the high-temperature firing stage, the formation of the liquid phase and gas generation in the ceramisite coarse aggregates serve as key factors in achieving burn swelling. The preparation of high-quality ceramisite coarse aggregates (CCAs) often requires complex composition design and high-precision heat treatment system control to achieve matching between the amount of generated liquid phase in the ceramisite coarse aggregates during the high-temperature firing stage, the viscosity and surface tension of the liquid phase, and the gas generation rate in the system.

Despite the severity of the “high-temperature foaming method”, a large amount of research on high-quality ceramisite was still based on this process. Tang et al. [3] used copper tailings, red mud, and fly ash as raw materials to prepare high-quality ceramisite coarse aggregates with high particle strength and a water absorption rate below 1%, by using the gas generated from the decomposition of Fe_2_O_3_ in red mud during high-temperature thermal treatment. Nguyen [4] investigated the impact of heat treatment temperature and insulation time on the structure and properties of ceramisite prepared from construction demolition waste and industrial byproducts, specifically, fly ash, with the objective of producing high-quality ceramisite coarse aggregates. Li [5] utilized sludge and waste glass powder as raw materials to prepare high-quality ceramisite. The resulting ceramisite exhibited a particle density of 0.78 g/cm^3^, cylindrical compression strength of 7.10 MPa, and water absorption rate of 1.01% after 1 h. Notably, the heat treatment temperature and insulation time were carefully controlled in this study.

Numerous scholars have effectively fabricated porous ceramisite with certain characteristics such as low density, low water absorption, and high strength in controlled laboratory settings [6,7,8,9]. However, these accomplishments relied heavily on an intricate composition design and precise temperature regulation. Therefore, implementing these techniques on a larger industrial scale remains a challenge.

Coal ash floating beads (CAFBs), derived from the combustion of coal, exhibit a spherical cavity structure and high melting point [10,11,12], with a widely tunable particle size within the range of 10–200 mesh. According to the above characteristics, in this study, a novel ceramisite preparation process was proposed that uses floating beads as the pore-forming templates and solid waste as raw materials. XRF, XRD, 3D micro-CT, and SEM were used to investigate the influence of floating bead particle size and addition amount on the phase composition, microstructure, and physical properties of ceramisite. The feasibility of using high-melting hollow spheres as templates for pore formation in the production of porous ceramisite were discussed.

## 2. Experimental Procedure

### 2.1. Raw Materials

Fly ash was obtained from Chuangtian Engineering Materials Co., Ltd. in Shijiangzhuang, China. CAFBs with different particle size were obtained from Lingshou County point gold mine product processing plant in Hebei, China. Molasses with solids of 48 wt% was obtained from Luying Chemical Co., Ltd. in Linyi, China. Table 1 and Table 2 present the specifications and chemical composition analysis of the raw materials, respectively. As can be seen, both of fly ash and CAFBs are mainly composed of SiO_2_ and Al_2_O_3_, followed by Fe_2_O_3_, CaO, and K_2_O, in addition to TiO_2_, MgO, and Na_2_O.

### 2.2. Methods

#### 2.2.1. Preparation of Ceramisite

In this experiment, fly ash was used as the raw material, and CAFBs with different particle sizes were used as the pore-forming templates. Ceramisite green bodies were prepared using a disc granulator. According to the experimental design, the raw materials were weighed and set aside. The disc granulator was then initiated, and a suitable amount of fly ash floating beads was introduced into the mixing pan. To create a homogeneous liquid film, a sugar syrup solution with a weight percentage of 15% was uniformly sprayed onto the surface of the float beads. Subsequently, an equivalent amount of fly ash powder was gradually incorporated, and additional sugar syrup solution was sprayed as necessary, depending on the coating condition of the powder on the float beads. This process was repeated in batches, with float beads and ceramisite powder added in equal proportions, until the ceramisite green bodies with particle sizes of 10–20 mm were obtained. After fully drying at 110 °C for 24 h, the ceramisite green bodies were heated in an experimental furnace at a rate of 10 °C/min until reaching 550 °C, and then held at this temperature for 1 h. The temperature was further increased at a rate of 5 °C/min until the desired temperature of 1200 °C was reached, and held at this temperature for 1.5 h. Afterward, the furnace was allowed to cool. Figure 1 illustrates the process flowchart for preparing ceramisite using floating beads as pore-forming templates.

From the preparation process of ceramisite in Figure 1, it can be found that the size of hollow floating beads will directly affect the size of pores in ceramisite, and the addition of floating beads will directly affect the volume fraction of pores in ceramisite. To investigate the impact of floating bead addition on the pore structure and properties of the ceramisite coarse aggregates, a ratio of 20-mesh floating beads to ceramisite raw materials was adjusted accordingly. The samples denoted as R5, R10, R15, and R20 corresponded to floating bead addition amounts of 5, 10, 15, and 20 wt%, respectively. Then, the influence of floating bead sizes on the structure and properties of ceramisite coarse aggregates was also examined by holding a constant mass ratio of fly ash powder to floating beads at 90:10. The samples labeled as R10-10, R10-20, R10-40, and R10-80 represent floating bead sizes of 10, 20, 40, and 80 mesh, respectively. The detailed experimental ratios are shown in Table 3.

#### 2.2.2. Testing and Characterization Methods

The mechanical properties of the fired ceramisite coarse aggregates, including cylinder compressive strength, 1 h water absorption rate, packing density, and bulk density were evaluated in compliance with the GB/17431.2-2010 standard [13]. In which, the cylinder compressive strength was tested using a universal material testing machine at a loading rate of 0.3–0.5 KN/s. The 1 h water absorption rate (*w*) was analyzed using Equation (1).
*w* = (*m_i_* − *m*_1_)/*m*_1_(1)
where *m*_1_ and *m_i_* are the dry weight of the ceramisite and the wet weight after immersion in water for 1 h, respectively. The apparent porosity (*P_a_*) and bulk density (*ρ_a_*) were tested using Archimedes’ water displacement method and calculated according to Equations (2) and (3).
*P_a_* = (*m*_3_ − *m*_1_)/(*m*_2_ − *m*_1_) × 100%(2)
*ρ_a_* = *m*_1_/(*m*_3_ − *m*_2_)(3)
where *m*_1_ and *m*_3_ are, respectively, the dry weight and the saturated wet weight of the ceramisite, and *m*_2_ is the suspended weight of the saturated ceramisite in water. In this work, *m*_1_ is equal to 100 g.

The true density of the ceramisite was determined using an automated density analyzer (AccuPyc 1340, Micromeritics, Norcross, GA, USA), while the phase composition of both the raw materials and the fired ceramisite was examined through X-ray diffraction analysis utilizing a Bruker D8 Advance instrument. The microstructural characteristics of the fired ceramisite coarse aggregates were observed with a TESCAN MIRA LMS scanning electron microscope, which was equipped with an energy dispersive spectrometer (Xplore, Oxford Instruments, Oxford, UK). Finally, the pore structure features of the fired ceramisite were investigated by means of industrial computed tomography (CT) technology provided by a GE Vtomex system.

## 3. Results and Discussion

### 3.1. Physical Properties

The results depicted in Figure 2 illustrate the impact of the size and quantity of the floating beads on the compressive strength of the ceramisite coarse aggregates. As shown in Figure 2a, when the addition amount of floating beads remained constant, the compressive strength of the ceramisite coarse aggregates initially increased and then decreased with decreasing size of floating beads. When floating beads had a size of 10 mesh, the compressive strength of the ceramisite coarse aggregates was 6.3 MPa. The compressive strength of the ceramic aggregates was maximized when the particle size of the floating beads was reduced to 40 mesh, reaching 12.5 MPa. Figure 2b shows the influence of floating bead addition on the compressive strength of the ceramisite coarse aggregates after firing. With an increase in the addition amount of floating beads, the compressive strength of the ceramisite coarse aggregates decreased. Notably, when the addition amount of floating beads was 20 wt%, the compressive strength of the ceramisite coarse aggregates was still as high as 8.7 MPa.

The loose/dense packing density, bulk density, and true density curves of the ceramisite coarse aggregates after firing, in relation to the quantity of floating beads added, are displayed in Figure 3a,b. As can be observed, the four density types of the ceramisite coarse aggregates after firing exhibited an approximately inverse proportional relationship with the addition amount of floating beads. This verified the feasibility of controlling the density level of the ceramisite coarse aggregates after firing by adjusting the addition amount of floating beads.

Figure 4a,b illustrates the graphical representations of the loose/dense bulk density, apparent density, and true density for the ceramisite coarse aggregates after firing. These measurements were taken with varying sizes of added floating beads, while maintaining the condition of a constant amount of floating beads added. The data revealed that: (1) a decrease in bead size correlated with an increase in both loose/dense bulk density and apparent density of the fired ceramisite coarse aggregates, and (2) the size of the beads had a negligible impact on the true density of the fired ceramisite coarse aggregates.

The results shown in Figure 2, Figure 3 and Figure 4 indicate that during the preparation stage of ceramisite raw materials, it is possible to regulate the apparent density and bulk density of the fired ceramisite by adjusting the amount and particle size of the added floating beads. Evidently, a higher proportion of floating beads leads to an increased porosity rate and decreased compressive strength of the fired ceramisite.

Ceramisite is a typical ceramic material, and the fracture strength can be analyzed using the Gibbs microcrack theory, as described in Equation (4).
(4)$$\sigma_{f}=\sqrt{2E\gamma_{S}/\pi C}$$

In this equation, σ*_f_* represents the critical fracture strength, *E* is the elastic modulus, *Υ_s_* is the fracture surface energy, and *C* is the semi-length of the pores (cracks). It is known that the material’s critical fracture strength is closely related to the size of the pores (cracks) within the material; clearly, the smaller the pore size (*C*), the higher the material’s critical strength tends to be.

Therefore, the analysis of the impact of floating beads size on the cylinder compressive strength of the fired ceramisite particles are as follows:

(1) Smaller floating bead particles contribute to a higher apparent density. With a consistent quantity of added floating beads, a reduction in bead size results in fewer overall pores within the ceramisite structure, consequently diminishing the total porosity of the ceramisite particles; (2) the size of the floating beads directly impacts the pore size within the ceramisite particles. When the total porosity remains comparable, finer pores within the material enhance the load-bearing capacity of the ceramisite particles. When floating beads had a size of 10 mesh, ceramisite particles had a minimum cylinder compressive strength; (3) decreasing floating bead size from 40 mesh to 80 mesh increases the likelihood of bead aggregation in the ceramisite raw materials. After high-temperature sintering, the pore structure formed by these aggregated beads emerges as the weak point in the ceramisite particles, thereby reducing the load-bearing capacity of the ceramisite particles.

Additionally, variations in packing density can influence the cylindrical compressive strength of the ceramisite particles. Evidently, a higher packing density enlarges the load-bearing area perpendicular to the compression direction, theoretically enhancing their compressive strength.

Figure 5 shows the relationship between the addition amount of floating beads (a) and the size of the floating beads (b) with the water absorption rate of the fired ceramisite coarse aggregates after 1 h. As can be observed, as the amount of floating beads increased, the water absorption rate of the fired ceramisite coarse aggregates slightly increased. However, when the amount of floating beads reached 20 wt%, the 1 h water absorption rate did not exceed 4.1%. With a decrease in the size of the floating beads, the 1 h water absorption rate of the ceramisite coarse aggregates initially decreased and reached its lowest value at 20 mesh, and subsequently presented a slight increase. With 10 wt% floating beads, and a mesh size of at least 20 mesh, the 1 h water absorption rate of the ceramisite coarse aggregates could be maintained at a relatively low level. Table 4 presents the physical properties of ceramisite coarse aggregates obtained and different reference values. As can be seen, the ceramisite coarse aggregates prepared in this work show significant advantages in low density, low water absorption rate, and high cylinder compressive strength, far exceeding the relevant provisions in GB/T17431.1-2010 in terms of properties. In addition, the self-produced ceramic particles also exhibit excellent advantages when compared to research findings by scholars worldwide. The results presented above indicated that the ceramisite produced by this process show significant advantages in properties such as cylinder compressive strength and water absorption rate.

### 3.2. CT Analysis

Figure 6 presents the 3D micro-CT reconstruction image of the ceramisite coarse aggregates after firing, with the colored regions in the image corresponding to the pores in the ceramisite coarse aggregates. As can be seen, the pores in the ceramisite coarse aggregates primarily exhibited a spherical structure, with an isolated distribution and relatively uniform dispersion. Additionally, the surface of the fired ceramisite coarse aggregates was relatively dense.

The 2D-CT reconstruction image shown in Figure 7 depicts a fired ceramisite particle, with the gray-white region representing the solid phase within the ceramisite particle, and the black region corresponding to the porous structure. Notably, the image revealed the presence of a dense shell layer on the outer surface of the ceramisite particle, effectively isolating the pores from the external environment. Numerous micron-sized spherical pores were also observed surrounding the larger-sized pores, which were derived from floating beads within the ceramisite particle. These micron-sized pores exhibited distinct and uniformly distributed structural characteristics, as demonstrated in Figure 7b.

### 3.3. Microstructure

Figure 8 shows the microscopic images of the floating beads, with Figure 8b depicting a magnified view of Figure 8a. As shown in the images, the floating beads exhibited a spherical structure. Additionally, numerous small-sized spherical floating beads with a particle size smaller than 10 μm were distributed on the surface and concave areas of the floating beads.

Figure 9 shows the typical microstructure images of ceramisite prepared using the floating bead template method, with Figure 9a,b depicting the 20-mesh and 80-mesh beads, respectively. The micrographs revealed that the pores in ceramisite exhibited a spherical structure and were isolated from each other. Moreover, certain pores within the coarse aggregates aligned with the dimensions of the floating beads, suggesting that the pore size in the ceramisite coarse aggregates could be regulated by manipulating the size of the floating beads.

Notably, the ceramisite coarse aggregates contained pores that fell into two distinct size categories. The first category consisted of pores originating from the initial floating beads that were introduced into the pores. These pores exhibited larger sizes and closely resembled the size of the initial floating beads, as indicated by the area marked in red in Figure 9. The second category comprised smaller pores that were predominantly spherical in shape, and these smaller pores were primarily located at the interfaces between the larger pores. Compared to the micropores presented in Figure 8, these spherical micropores had a similar size and shape to the spherical hollow microbeads attached to the surfaces of the initial floating beads. It can be speculated that during the preparation stage of the ceramisite green bodies, the spherical hollow microbeads attached to the surface of the initial floating beads became detached from the beads under the action of frictional force and evenly dispersed into the ceramisite green bodies.

### 3.4. Phase Analysis

Figure 10 presents the phase analysis of the fly ash, floating beads, and different amounts of floating beads added to the ceramisite coarse aggregates. It was observed that the phase composition across all samples was fundamentally identical, and primarily composed of mullite, quartz, and Al_1.2_Ca_0.2_Na_0.8_O_8_Si_2.8_. However, compared to the fly ash and floating beads, the diffraction peak intensity of mullite in the phase composition of the ceramisite R5 and R20 particle samples after firing significantly increased. As can be seen, the diffraction peaks of mullite, situated between 26 and 28 degrees, demonstrated a particularly pronounced variation. Moreover, with an increase in the amount of added floating beads, the pronounced enhancement of mullite diffraction peaks were also observed in the phase composition of the ceramisite coarse aggregates.

The experimental results demonstrated that ceramisite coarse aggregates could be derived from fly ash as the primary material, and floating beads as the pore-forming templates. After high-temperature sintering, these granules, in contrast to the lightweight high-strength ceramisite granules described in GB/T17431.1-2010, exhibited a remarkable enhancement in bulk compressive strength and a notable decrease in water absorption while maintaining the same bulk density level.

### 3.5. Analysis of the Adjustable and Controllable Structure of the Ceramisite Coarse Aggregates

The floating beads had a spherical cavity structure, which provided a template selection for the pore structure construction of lightweight and high-strength ceramisite coarse aggregates. The floating beads possessed a high melting point, which ensured the stability of the pore structure during high-temperature sintering of the ceramisite coarse aggregates. Moreover, a similarity in composition between the fly ash and floating beads allowed the cavity structure of the beads to resist erosion caused by the low melting point phases that formed during high-temperature sintering.

Based on the above three factors, the manipulation and design of the pore structure parameters of the fired ceramisite coarse aggregates could be achieved by modifying the size and quantity, and utilized for introducing floating beads, especially during the initial stage of ceramisite green body preparation.

### 3.6. Analysis of the Simultaneous Consideration of Light Weight and Strength

#### Characteristics of the Ceramisite Coarse Aggregates

The pores in the ceramisite coarse aggregates after firing mainly arose from the coal ash beads used as the pore-forming template. These pores were mostly in the form of spherical structures and were isolated from each other, and this characteristic effectively delayed the occurrence of stress concentrations within the ceramisite particle structures when subjected to load conditions. Consequently, this enhanced the load-bearing capacity of the ceramisite coarse aggregates, allowing them to maintain a significantly higher cylinder pressure strength than the national standard, while still maintaining a light weight.

Figure 11a presents the microstructure image of the derived pores originating from the typical beads in the fired ceramisite coarse aggregates, where Figure 11b depicts the energy spectrum analysis results at point 1 in Figure 11a. The inner wall of the pores exhibited numerous whisker-like structures, and energy spectrum analysis revealed that the primary elemental composition at point 1 was aluminum (Al), silicon (Si), and oxygen (O). According to the phase analysis results illustrated in Figure 6, we inferred that the whisker-like phase situated on the inner walls of the pores corresponded to mullite. The presence of these whisker-like mullite structures contributed to the mechanical strength enhancement of the ceramisite coarse aggregates.

### 3.7. Analysis of the Low Water Absorption Rate of Ceramisite Granules

Figure 12 illustrates the effects of fly ash floating bead particle size on the total porosity and closed pore ratio of the ceramisite coarse aggregates after firing, with 10 wt% addition. As can be observed, as the size of the pearls decreased, both the total porosity and closed pore ratio of the ceramisite granules decreased. Moreover, the ratio of closed pore ratio to total porosity in all samples exceeded 89%, indicating that the closed pores were the dominant pore type in the ceramisite coarse aggregates.

Combined with the CT results shown in Figure 2 and Figure 3, as well as the microstructure of the fired ceramisite coarse aggregates (Figure 5), the pores in the ceramisite coarse aggregates were generally distributed in an isolated manner. Additionally, a dense shell layer was observed on the outer surface of the fired ceramisite coarse aggregates. The combination of these factors resulted in a very low water absorption rate of the fired ceramisite coarse aggregates.

## 4. Conclusions

By using fly ash as the raw material and CAFBs as the pore-forming templates, ceramisite green bodies were prepared using a disc granulator. After high-temperature sintering, lightweight, high-strength, and low water absorption ceramisite coarse aggregates with excellent performance indicators were obtained, which far exceed the lightweight and high-strength requirements of the ceramisite coarse aggregates described in GB/T 17431.1-2010. With the addition of 40-mesh CAFBs at 10 wt%, high-quality ceramisite coarse aggregates with a packing density of 823 kg/m^3^, a cylinder compressive strength of 12.5 MPa, and a 1 h water absorption rate of 4.0% were successfully produced.The pore structure characteristics of the ceramisite coarse aggregates after firing could be tailored and regulated through the manipulation of floating bead size, quantity, and distribution.The resulting ceramisite coarse aggregates possessed a spherical isolated distribution of pores and a dense shell layer on the surface after the firing process. This structural arrangement provided a structural basis for the preparation of high-quality ceramisite coarse aggregates.

## Figures and Tables

**Figure 1 materials-17-02613-f001:**
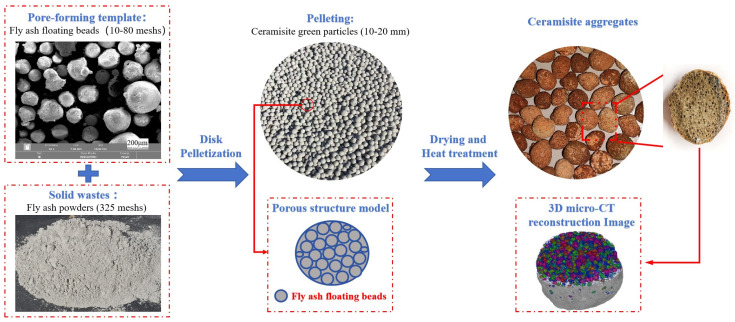
New preparation process of lightweight and high-strength ceramisite.

**Figure 2 materials-17-02613-f002:**
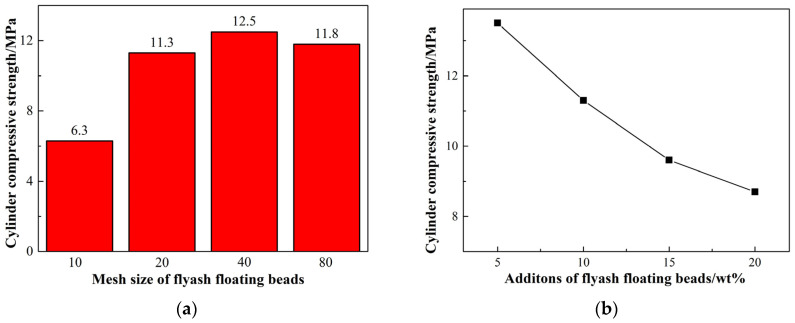
Effects of particle size (**a**) and addition amount (**b**) of CAFBs on the cylinder compressive strength of ceramisite.

**Figure 3 materials-17-02613-f003:**
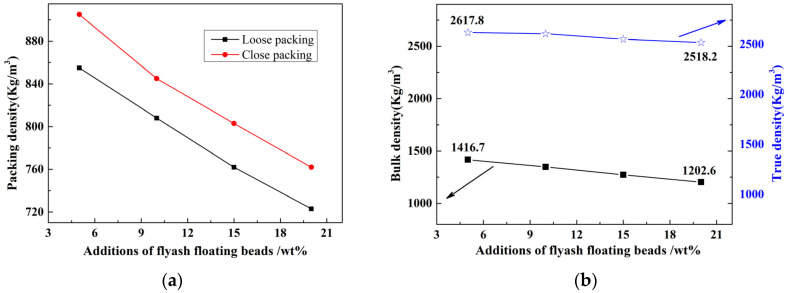
Effects of CAFB addition on the packing density (**a**), as well as bulk density and true density (**b**) of ceramisite.

**Figure 4 materials-17-02613-f004:**
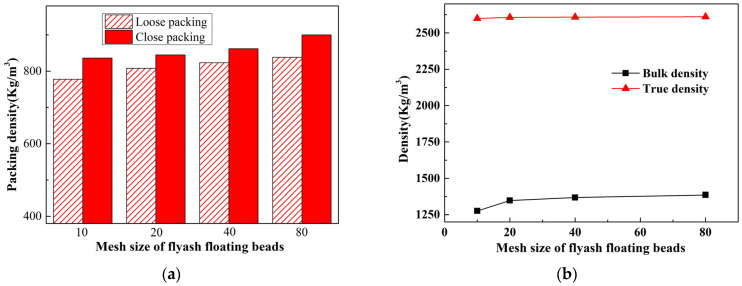
Effects of CAFB particle size on the packing density (**a**), as well as bulk density and true density (**b**) of ceramisite.

**Figure 5 materials-17-02613-f005:**
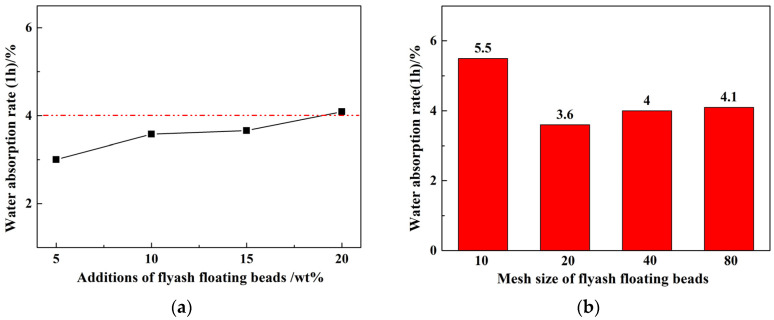
Effects of particle size (**a**) and addition amount (**b**) of CAFBs on the 1 h water rate absorption of ceramisite.

**Figure 6 materials-17-02613-f006:**
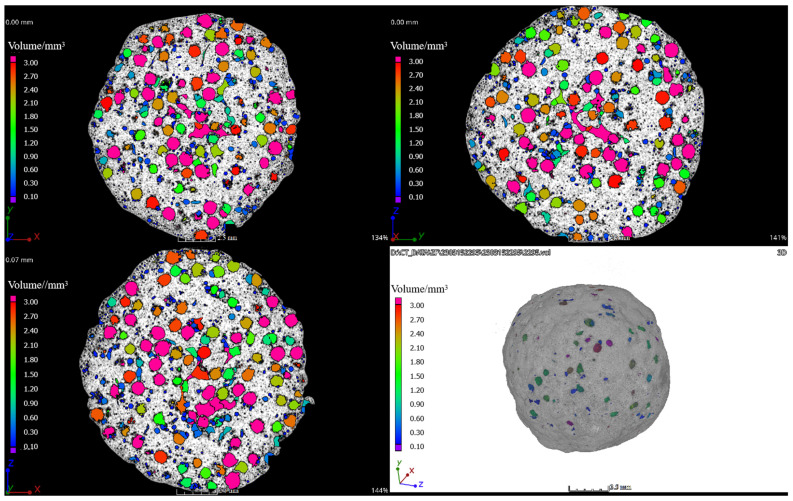
Three-dimensional micro-CT reconstruction images of the R15 sintered ceramisite sample.

**Figure 7 materials-17-02613-f007:**
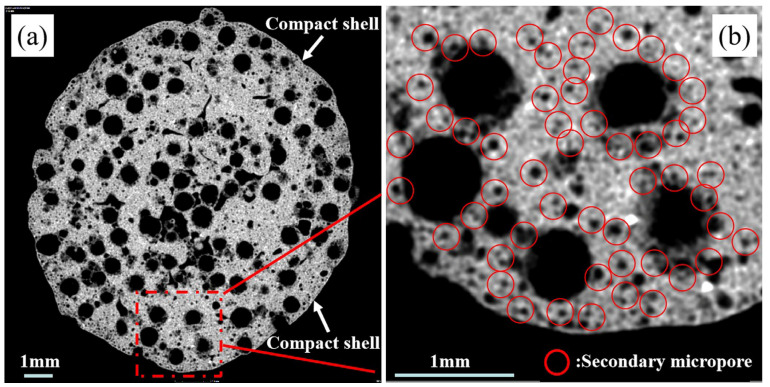
Two-dimensional micro-CT reconstruction picture of sintered ceramisite sample R15 (**a**), where (**b**) shows a partially enlarged drawing of (**a**).

**Figure 8 materials-17-02613-f008:**
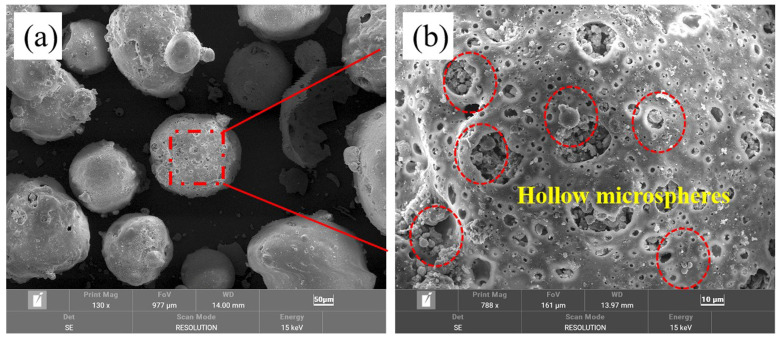
Microstructure of the CAFB (**a**), and (**b**) partially enlarged image of (**a**).

**Figure 9 materials-17-02613-f009:**
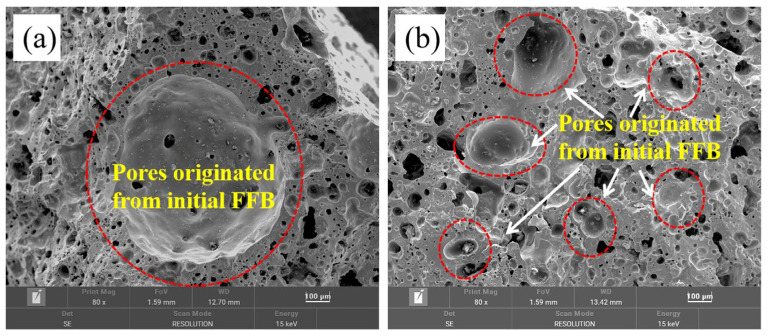
Typical microstructures of the CAFB: (**a**) 20 mesh and (**b**) 80 mesh.

**Figure 10 materials-17-02613-f010:**
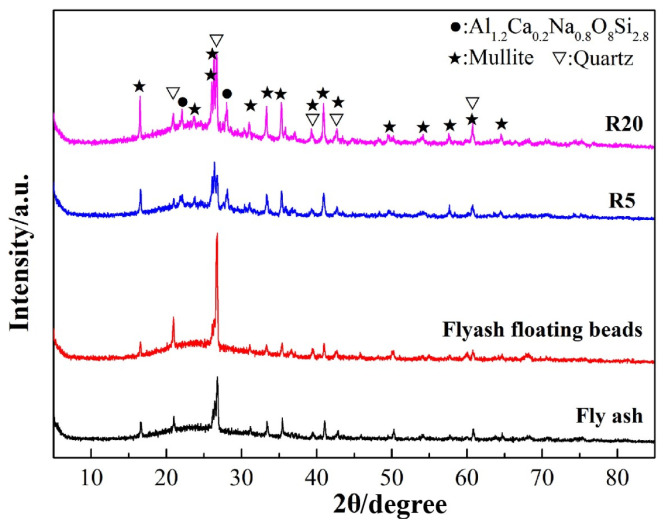
Phase analysis of the raw materials and sintered ceramisite R5 and R20 samples.

**Figure 11 materials-17-02613-f011:**
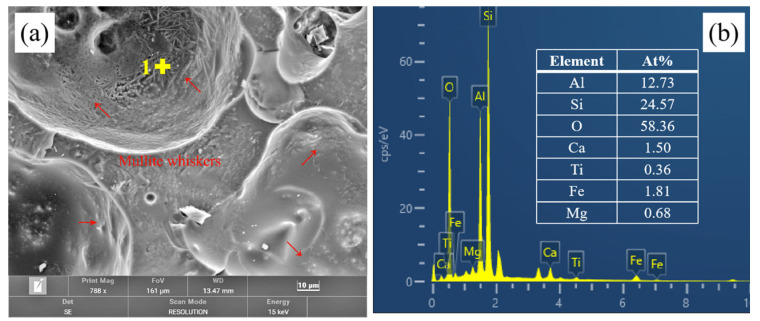
Microstructure image of the typical floating bead-derived pores in ceramisite (**a**,**b**) EDS analysis of point 1 in (**a**).

**Figure 12 materials-17-02613-f012:**
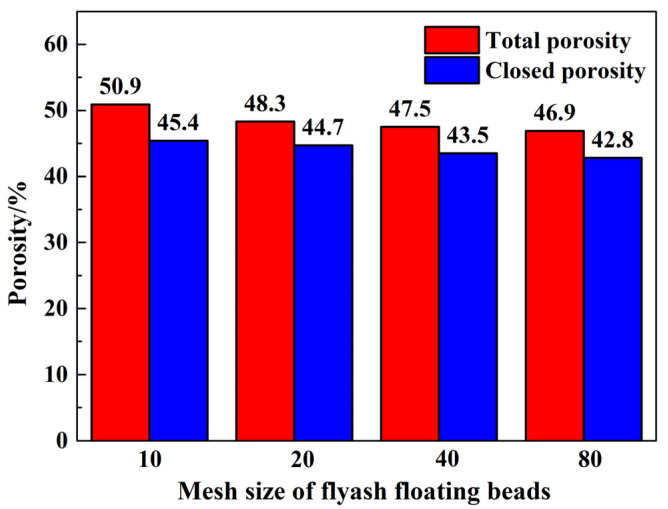
Effects of CAFB particle size on the total porosity and closed porosity of the ceramisite.

**Table 1 materials-17-02613-t001:** Specification parameters of the raw materials.

Raw Materials	Specification Parameters
Fly ash	325 mesh
Coal ash floating beads (CAFBs)	10, 20, 40, 80 mesh
Molasses	Solid content of 48 wt%

**Table 2 materials-17-02613-t002:** Chemical composition of the raw materials.

Raw Materials	wt%									
	SiO_2_	Al_2_O_3_	Fe_2_O_3_	CaO	K_2_O	TiO_2_	MgO	Na_2_O	P_2_O_5_	Loss
Fly ash	52.26	30.18	6.60	3.69	2.76	1.31	0.80	0.70	0.33	1.36
Coal ash floating beads (CAFBs)	59.02	26.43	5.55	1.53	2.83	0.98	1.30	1.73	0.20	0.43

**Table 3 materials-17-02613-t003:** Experimental design.

Sample Number	Fly Ash (wt%)	CAFB (wt%)			
		10 Mesh	20 Mesh	40 Mesh	80 Mesh
R5	95		5		
R10-10	90	10			
R10-20	90		10		
R10-40	90			10	
R10-80	90				10
R15	85		15		
R20	80		20		

**Table 4 materials-17-02613-t004:** Physical properties of the as-prepared ceramisite particles and different reference values.

Process Type	Physical Properties of Ceramisite							Reference
	Loose Packing Density /(kg·m^−3^)	Bulk Density /(g·cm^−3^)	1 h Water Absorption Rate/%	24 h Water Absorption Rate/%	CCS /MPa	Porosity /%	Particle Size/mm	
High Melting Point Hollow Sphere Pore-Forming Template Process	778–838	1.275–1.385	4.10–5.50	--	6.3–11.8	46.9–50.9	10–20	This work
	723–855	1.202–1.416	3.00–4.09	--	8.7–13.5	45.9–52.2	10–20	
	500–600	--	≤10	--	4.0	--	10–20	GB/T17431.1-2010 [14]
	600–700	--	≤10	--	5.0	--	10–20	
	700–800	--	≤10	--	6.0	--	10–20	
	800–900	--	≤10	--	6.5	--	10–20	
Sintering Process	386–507	--	--	--	1.11–6.30	46.0–67.5	5–15	Wang et al. (2022) [15]
	630–730	1.24–1.48	--	23.54–38.36	1.23–3.01		8–10	Beatriz González-Corrochano et al. (2016) [9]
High-Temperature Foaming Method	570–820	--	1.01–1.65	--	6.75–11.51	--	10	Tang et al. (2023) [3]
	550–990	--	10–16	--	1–8	--	6–10	Hung Phong Nguyen. (2021) [4]
	About 580–1023	--	5–42	--	1.52–8.24	--	--	Han et al.(2021) [1]
	780	--	1.01	--	7.1	54.91	15 ± 1	Li et al. (2021) [5]
	--	0.755–1.268	50–14	--	0.5–6.18	--	10–15	Cao et al. (2019) [6]
	--	0.862–1.300	36–4	--	2.02–8.82	--	10–15	Cao et al. (2019) [6]
	836	1.672	<12	--	13.7	--	5–8	Liu et al. (2018) [7]
	746.6–982.4	1.527–1.814	6.7–16.53	--	4.25–10.53	--	5–8	Li et al. (2020) [16]
	280	--	12.5	--	1.0	--	12	Pei et al. (2022) [17]
	410–995	0.775–1.569	0.3–3.5	--	1.8–12.4	--	12	Jiang et al. (2023) [18]
	640–1071	About 1.0–1.75	2.4–13.5	--	3.3–9.0	--	8–10	Li et al. (2023) [19]
	810	1.19	3.5	--	11.2	--	--	Long et al. (2023) [20]

## Data Availability

Data are contained within the article.

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
