# Peer review of "A Novel Method for Preparing Lightweight and High-Strength Ceramisite Coarse Aggregates from Solid Waste Materials"

_materials, 2024, doi:10.3390/ma17112613_

Round 1

Reviewer 1 Report

Comments and Suggestions for Authors

The presented research issues regarding selected structural, strength parameters and a diagnostic method based on computed tomography for ceramisite coarse aggregates are scientifically interesting and currently show great demand for the industry. Below are some comments and suggestions:

1. The introduction should expand the information regarding ceramsite laboratory tests using XRF and other methods - so that the introduction constitutes a background to previous tests;

2. In the subsection 2.2, please write whether the preparation of ceramsite was carried out according to any procedure or standard;

3. In the subsection 2.2.2, write how many samples were prepared for testing;

4. In the subsection 3.1, please add information regarding the minimum value of the compressive strength of the ceramsite coarse aggregates. In addition, please add a few sentences regarding the relationship between packing density and compressive strength;

5. In the subsection 3.4 for Figure 10, it should be written more clearly what the intensities mean at the value between 26-28 on the horizontal axis. Additionally, the full name must be added on the horizontal axis;

6. Line 334 – no Figure attached;

7. The conclusions should include two/three sentences regarding the numerical results from laboratory tests, in particular in terms of structural and strength parameters.

Author Response

  1. The introduction should expand the information regarding ceramsite laboratory tests using XRF and other methods - so that the introduction constitutes a background to previous tests;

Response:This has been supplemented in the introduction, as detailed in lines 65-71.

  1. In the subsection 2.2, please write whether the preparation of ceramsite was carried out according to any procedure or standard;

Response:To my knowledge, the use of high-melting-point hollow spheres as pore-forming templates for the preparation of ceramisite is an original work of this study, which has been detailed in Section 2.2.1.

  1. In the subsection 2.2.2, write how many samples were prepared for testing;

Response:The supplement has been made in accordance with the review comments, as detailed in lines 131-132.

  1. In the subsection 3.1, please add information regarding the minimum value of the compressive strength of the ceramsite coarse aggregates. In addition, please add a few sentences regarding the relationship between packing density and compressive strength;

Response:The supplement has been made in accordance with the review comments, as detailed in lines 147-150; 197-198 and 203-207.

  1. In the subsection 3.4 for Figure 10, it should be written more clearly what the intensities mean at the value between 26-28 on the horizontal axis. Additionally, the full name must be added on the horizontal axis;

Response:The supplement has been made in accordance with the review comments, as detailed in lines 284-286. Additionally, Figure 10 has been revised.

  1. Line 334 – no Figure attached;

Response:The correction has been completed.

7.The conclusions should include two/three sentences regarding the numerical results from laboratory tests, in particular in terms of structural and strength parameters.

Response:The supplement has been made in accordance with the review comments, as detailed in lines 355-358.

Reviewer 2 Report

Comments and Suggestions for Authors

From my point of view, this text could become an interesting article. However, in my opinion, this text is not acceptable to be published yet due to the following.

1- Although the abstract is very clear and well-written the rest of the article is too confusing for potential readers to understand the work presented by the authors.

2- The main weakness is the confusion regarding the scope, objectives, aim… of the article. Another important weakness is the organization, structure of the article. Both lacks imply that a general improvement is required.

3- Regarding the aim, scope, objectives, boundaries…, these should be more clearly defined and coherent all through the article. In this version, potential readers would understand from some parts of the text that a new model for producing ceramisite coarse aggregate is developed and presented because the main gap in the former technical literature is implementing these techniques to a larger industrial scale. On the other hand, the article is at the end a presentation of an experimental campaign that requires a better organization. In consequence, the aim, scope, objectives, boundaries…, need to be further defined and coherent in all parts, from abstract to conclusions.

Other comments

4- The introduction requires rewriting. The first to paragraphs present strong statements but lack references or evidence to rely on. The next paragraph is not sufficient to properly present the context and the former related projects. The fourth paragraph is very confusing in presenting the gap in the technical literature that this research paper covers. The objectives and topic are also unclear, vague and unjustified, for example the sentence in lines 59-60. The final paragraph in the introduction is summarizing findings and, therefore, is not well located. The summary of the article is required in the abstract while the main findings are required in the conclusions. Instead, the introduction lacks some final lines in the introduction describing briefly the main sections and subsections of the article and its contents.

5- The results and discussion section requires improvement in presenting its contents. To give a couple of examples, the comparision of results between this project, the standards and previous literature should be discussed in the text, and explain why the last line has values in blue. The fact that section 3 is results and discussion and subsection 3.5 is discussion is uncommon and should be explained to potential readers.

6-There are some expressions that diminish the rigor of the paper and are inadequate for a scientific article, p.e. “we” (line 65; line 157; line 336, etc.).

7- Check the Names of figures through the text, there are unexplained isolated references to figures such as Fig.4 in Line 162, and Fig.12 in Line 334

8- Review the citing format of the references numbers within the text manuscript

Author Response

Reviewer #2:

1- Although the abstract is very clear and well-written the rest of the article is too confusing for potential readers to understand the work presented by the authors.

Response:Thank you very much for your review comments; we have made appropriate adjustments to the manuscript in accordance with your suggestions. It should be explained that the fundamental structure of this manuscript is based on the writing style found in journals such as Construction and Building Materials, Journal of Cleaner Production, and Ceramics International.

2- The main weakness is the confusion regarding the scope, objectives, aim… of the article. Another important weakness is the organization, structure of the article. Both lacks imply that a general improvement is required.

Response:Thank you very much for your review comments; we have made appropriate adjustments to the manuscript in accordance with your suggestions.

  • Regarding the aim, scope, objectives, boundaries…, these should be more clearly defined and coherent all through the article. In this version, potential readers would understand from some parts of the text that a new model for producing ceramisite coarse aggregate is developed and presented because the main gap in the former technical literature is implementing these techniques to a larger industrial scale. On the other hand, the article is at the end a presentation of an experimental campaign that requires a better organization. In consequence, the aim, scope, objectives, boundaries…, need to be further defined and coherent in all parts, from abstract to conclusions.

Response:We deeply appreciate the reviewer’s suggestion.According to the reviewer’s comment, we have made appropriate adjustments to the manuscript in accordance with your suggestions. It should be explained that the fundamental structure of this manuscript is based on the writing style found in journals such as Construction and Building Materials, Journal of Cleaner Production, and Ceramics International.

Other comments

  • The introduction requires rewriting. The first to paragraphs present strong statements but lack references or evidence to rely on. The next paragraph is not sufficient to properly present the context and the former related projects. The fourth paragraph is very confusing in presenting the gap in the technical literature that this research paper covers. The objectives and topic are also unclear, vague and unjustified, for example the sentence in lines 59-60. The final paragraph in the introduction is summarizing findings and, therefore, is not well located. The summary of the article is required in the abstract while the main findings are required in the conclusions. Instead, the introduction lacks some final lines in the introduction describing briefly the main sections and subsections of the article and its contents.

Response:Revisions have been made according to the review comments, as detailed in the highlighted sections of the introduction.

  • The results and discussion section requires improvement in presenting its contents. To give a couple of examples, the comparision of results between this project, the standards and previous literature should be discussed in the text, and explain why the last line has values in blue. The fact that section 3 is results and discussion and subsection 3.5 is discussion is uncommon and should be explained to potential readers.

Response:Thank you very much for your review comments. We have made the appropriate modifications and corrections in accordance with your suggestions. The details are on pages 225-227; section 3.5 has also been adjusted accordingly.

  • There are some expressions that diminish the rigor of the paper and are inadequate for a scientific article, p.e. “we” (line 65; line 157; line 336, etc.).

Response:Modifications and corrections have been made in accordance with the review comments. See lines 63, 211, 283, 287, and 338 for details.

7- Check the Names of figures through the text, there are unexplained isolated references to figures such as Fig.4 in Line 162, and Fig.12 in Line 334

Response:The correction has been completed.

  • Review the citing format of the references numbers within the text manuscript

Response:The correction has been completed.

Reviewer 3 Report

Comments and Suggestions for Authors

The manuscript deals with an interesting research topic but there is room for organizational  improvements and argumentation upgrading, thus, the following review comments can be considered prior to acceptance for publication at the Materials journal.

-It seems that the order of citations is random, therefore, it has to be reordered from 1,2,3,……., not from citation 3 and then the missing citations 13-20 have to be also added/noted in the relevant text.

-Sections 2 and 3 are almost all deprived from citations, thus, checking and cross-citing the relevant analysis and findings can be better validated and verified in alignment with those relevant from other authors’ citations.

-The bold-typing of Table 4 has to change to normal-plain text, except for the Table 4-heading-line.

-In line 78 the “3” has to be positioned as superscript, in cm^3.

-The technological background of the matierial synthesis has been fully developed. However, due to the procedural demand that “…..550°C, and then held at this temperature for 1 h. The temperature was further increased at a rate of 5°C/min until the desired temperature of 1200°C was reached, and held at this temperature for 1.5 h…”, it is implied an energy-intensive process, thus, the a) energy restrictions, b) environmental and safety concerns, have to be succinctly discussed as separate, autonomous, subsection within the main Discussion section. For this, 2-3 and cross-cited paragraphs are adequate.

Author Response

Reviewer #3:

  • It seems that the order of citations is random, therefore, it has to be reordered from 1,2,3,……., not from citation 3 and then the missing citations 13-20 have to be also added/noted in the relevant text.

Response:Thank you very much for your review comments. References 1-3 have been added to the introduction, and references 13-20 are marked in Table 4.

  • Sections 2 and 3 are almost all deprived from citations, thus, checking and cross-citing the relevant analysis and findings can be better validated and verified in alignment with those relevant from other authors’

Response:Thank you very much for your review comments. The preparation process of ceramisite described in Section 2 is proposed for the first time in this work. Subsequently, the performance testing of the ceramisite primarily refers to the GB/17431.2-2010 standard. Due to the above reasons, no references are cited in Section 2 of this article. In Section 3, Table 4 cites performance data from ceramisite research reports by scholars worldwide and compares them.

  • The bold-typing of Table 4 has to change to normal-plain text, except for the Table 4-heading-line.

Response:Modifications have been made in accordance with the reviewer's comments, as detailed in Table 4.

  • In line 78 the “3” has to be positioned as superscript, in cm^3.

Response:Modifications have been made in accordance with the reviewer's comments.

  • The technological background of the matierial synthesis has been fully developed. However, due to the procedural demand that “…..550°C, and then held at this temperature for 1 h. The temperature was further increased at a rate of 5°C/min until the desired temperature of 1200°C was reached, and held at this temperature for 1.5 h…”, it is implied an energy-intensive process, thus, the a) energy restrictions, b) environmental and safety concerns, have to be succinctly discussed as separate, autonomous, subsection within the main Discussion section. For this, 2-3 and cross-cited paragraphs are adequate.

Response:High-temperature treatment is the most effective method for producing lightweight and high-strength ceramisite. With the increasing global requirements for environmental protection and carbon emissions, many ceramisite manufacturing enterprises in China are now equipped with a very comprehensive set of equipment for exhaust gas treatment, wastewater disposal, and waste heat power generation.

Round 2

Reviewer 2 Report

Comments and Suggestions for Authors

This article has been improved following the eviewers comments and, after adapting the references to the journal format and reviewing the typewriting of the text it will be ready for publication

Comments on the Quality of English Language

There are typewriting errors through the text that must be fixed. For example, line 205 "cordingly.."

Author Response

Dear Editors and reviewers:

Thank you for your precious comments and advice. We have comprehensively corrected the spelling and grammar of all words in this manuscript. Thank you very much for your patient and professional review.